# Built-In Self-Test (BIST) Methods for MEMS: A Review

**DOI:** 10.3390/mi12010040

**Published:** 2020-12-31

**Authors:** Gergely Hantos, David Flynn, Marc P. Y. Desmulliez

**Affiliations:** Smart Systems Group, Earl Mountbatten Building, Research Institute of Sensors, Signals and Systems, School of Engineering & Physical Sciences, Heriot-Watt University, Edinburgh EH14 4AS, UK; D.Flynn@hw.ac.uk (D.F.); M.Desmulliez@hw.ac.uk (M.P.Y.D.)

**Keywords:** micro-electro-mechanical systems (MEMS) test, built-in-self-test (BIST), failure modes, multi-functional sensors

## Abstract

A novel taxonomy of built-in self-test (BIST) methods is presented for the testing of micro-electro-mechanical systems (MEMS). With MEMS testing representing 50% of the total costs of the end product, BIST solutions that are cost-effective, non-intrusive and able to operate non-intrusively during system operation are being actively sought after. After an extensive review of the various testing methods, a classification table is provided that benchmarks such methods according to four performance metrics: ease of implementation, usefulness, test duration and power consumption. The performance table provides also the domain of application of the method that includes field test, power-on test or assembly phase test. Although BIST methods are application dependent, the use of the inherent multi-modal sensing capability of most sensors offers interesting prospects for effective BIST, as well as built-in self-repair (BISR).

## 1. Introduction

The earliest micro-electro-mechanical-systems (MEMS) emerged in the 1970s [1] and interest in creating them has grown ever since. Unlike in integrated circuits (ICs), MEMS utilise information and material from multiple domains of energy (electrical, mechanical, thermal, optical, biological, etc.), making testing in general, and built-in self-test (BIST) for MEMS, much more complicated than for ICs. Moreover, the causes and modes of failures are much more numerous and varied, rendering online and offline testing complicated [2,3]. In comparison to IC testing, MEMS testing is therefore more expensive generally, representing, in some cases, up to 50% of the price of the end product [4,5,6]; built-in self-test (BIST) methods can help offset these cost-provided solutions are offered to test across multiple domains of energy and interfaces. However, due to the increased complexity as compared to ICs, BIST for MEMS only emerged in 1989 [7].

BIST enables an electronic system to be aware of its own condition [8]. This technique found ubiquitously in embedded systems allows the extraction of parameters ranging from a single system pass/fail information to a comprehensive set of parametric information regarding the health of each of the system components. Using the terminology employed by Benso et al. [9], BIST is a “detection only” fault handling strategy, whereby the system signals only the detection of a fault, but does not act on it and lets the external environment handle it.

First developed for integrated circuits (ICs) in 1977 [10], and considered a well-established design-for-testability (DfT) method in the 1980s [11], BIST has evolved into various implementation specific techniques, which include histogram-based analog BIST (HABIST) [12], logic BIST (LBIST) [13], memory BIST (MBIST) [14], and its programmable alternative, programmable BIST (PBIST) [15]. BIST is also used alongside various tools for testing, such as JTAG [16], automatic test pattern generation (ATPG) [17], for example, or sometimes incorporating tools such as cyclic redundancy check (CRC) [18] or pseudo-random pattern generator (PRPG) [19]. MBIST is also usually used in conjunction with redundancy-based built-in self-repair (BISR) or memory built-in self-repair (MBISR), techniques which implement the complete removal of the fault [20,21] and generally rely on the redundancy of electronic components [22].

This article presents a systematic review and classification of BIST strategies for MEMS. A moderate amount of literature is available on BIST techniques for MEMS, with all articles presenting solutions specific to the MEMS studied, and most of them focusing on inertial MEMS [23,24]. As the last review on BIST for MEMS was written in 2006 [25], this review article provides a timely up-to-date overview based on around 100 publications of academic and industrial research worldwide.

A taxonomy of the various BIST strategies for MEMS discussed in this review is presented in Figure 1. Section 2 introduces electrically induced stimuli test methods. These methods are the most popular and have their origin in the testing of inertial MEMS. In Section 3, the authors present a set of test methods which they define as delay-based test methods; these form the second most popular test methods. Section 4 discusses the impulse response test methods and Section 5 describes the bias superposition/modulation methods, which are becoming very popular. Section 6 highlights the unique use of sensors with different sensing modalities at a potential BIST method. At the end of each section, a discussion paragraph summarises the advantages and drawbacks of each methods alongside their key characteristics. Section 7 benchmarks all these methods in terms of ease of implementation, “usefulness” defined as the number of functionalities a given method possesses, and test time duration with indication as whether the test is occurring during manufacturing/assembly, at activation or during operation of the device. A summary table highlighting the key performance metrics for each BIST method is also provided. Finally, Section 8 provides general conclusions regarding this review article.

## 2. Electrically Induced Stimuli Test

In this test method, an electrical signal generates a physical signal, like an electrostatic force for example, inside the MEMS. This physical signal is expected to generate a response from key functional sub-structures of the MEMS. If the response can be converted back into an electrical signal, a viable BIST method would use this physical signal either to mimic the physical forces acting on the device under test (DUT) under normal operating conditions, or to extract information about the health of the DUT [23,24]. Four test strategies that highlight this method are presented in this subsection: single-ended test, symmetrical test, direct and indirect parameter extraction.

### 2.1. Single-Ended Test

Many MEMS have moving or free-standing structures that either (1) have been designed purposely to be actuated, (2) are susceptible to be deformed, albeit unintentionally, or (3) can be used by some other means to stimulate other parts. Examples include the use of electrostatic forces for accelerometer testing [26], the heating of a resistor to cause thermal expansion of a suspended structure [27,28], or the emission of infrared signals impinging on a thermopile structure [29]. An electrical test signal generator would electrically stimulate the device in a way that results in a physical movement, a deformation, or the creation of a signal belonging to the right domain of energy for the device to be tested. The output response is then measured and compared to the normal operational behaviour of the system [30]. This subsection presents in detail these three possibilities.

#### 2.1.1. Electrostatic Actuation

Developed as early as 1989 for crash sensors in car airbags, the method relies on the generation of an electrostatic force to actuate the freestanding seismic mass of a capacitive accelerometer. An example of this self-test system can be found in the former ADXL50 acceleration measurement system commercialised by Analog Devices [26]. It not only supports self-test but also self-calibration using the Earth’s gravity.

Zimmerman et al. present also an accelerometer structure with self-test capability of duration of less than 2 ms, utilizing electrostatic force to deflect the moving mass as shown in Figure 2 [31]. The generated pulse signal makes the accelerometer behave identically to an external acceleration signal. The amplitude of the deflection is controllable. The self-test is claimed to be useful also for the periodic calibration and accelerated ageing test. Analysis of the test response allows access to the functional parameters of the device, such as the resonance frequency, which is affected by most common accelerometer faults [32,33].

#### 2.1.2. Thermal Actuation

Thermal actuation is utilized for BIST in various ways [34,35]. In one instance, a current is used to thermally expand a bimetallic membrane to simulate membrane deflection in a pressure sensor [27,28]. Charlot et al. use thermal actuation in three applications: (1) a cantilever structure used as a tactile sensor, (2) a cantilever with a suspended mass simulating an accelerometer and a (3) membrane for pressure sensors [24]. In all three cases, a heater electrode is used to provoke the thermally induced deformation of a suspended structure. The magnitude of the deformation is detected by piezoresistors, as shown in Figure 3.

#### 2.1.3. Other Thermally Induced Effects

Heat can also be used indirectly to actuate free-standing structures. One example, shown in Figure 4, includes the heating of air present in a cavity, which generates a pressure gradient that causes the deflection of a membrane in a MEMS pressure sensor [36]. Heater electrodes are also used to provide a built-in heat source for a MEMS infrared imager [23]. In all cases, the thermal element is a simple heating electrode with a meander structure implemented on a chip.

In summary, a singe-ended test has been demonstrated for a variety of MEMS, including accelerometers, tactile and pressure sensors, and infrared imagers. Each application has different time constraints ranging from less than 2 ms in airbag application using capacitive accelerometers [31] to around 3 ms for the thermal excitation of the cantilever-type accelerometer structure. The main benefit of this test method is the lack of need for a physical force to run the measurement. The method is also relatively fast, but application-dependent. Additional integrated circuitry or external readout equipment is however required to analyse the sensor output.

### 2.2. Symmetrical Test Method

Similar to the single-ended test method in terms of the generation of the physical stimuli, symmetrical testing focuses on enhancing the observability of deviations of device response from normal behaviour. This BIST method relies on detecting any differences in the results obtained at symmetrically placed measurement points and resulting from the presence of local defects. This test, which results in pass/fail information about the health of the device, is not effective for defects such as stiction, that affects the device in a symmetric manner, but detects finger height mismatch and etch variation very well in the case of a comb-like accelerometer.

Most of the research related to this method has been carried out at Carnegie Mellon University by Deb et al. [37,38,39,40], and by Xiong at al. at the University of Cincinnati, as shown in Figure 5 [41,42]. While Deb et al. focused on the symmetrical test method only on surface-micromachined capacitive MEMS accelerometers, Xiong et al. showed applied both single-ended and symmetrical test for three different devices and claimed that their technique can be easily extended to other capacitive MEMS devices. With a 1 MHz modulation voltage, the sensing time is under 1 ms. This test method also requires additional circuitry for control and measurement at symmetrically placed areas.

### 2.3. Direct Parameter Extraction Test

Electrical parameters obtained from measurements of the output response of MEMS very often show a strong correlation with their mechanical parameters (mass, damping coefficient, spring constant). It would be therefore possible to directly infer from these measurements, whether the mechanical structure of MEMS is affected by defects. In that regard, Natarajan et al. came up with a method to test certain mechanical parameters of a capacitive accelerometer by purely electrical means [43]. A large frequency AC signal was used to create an oscillating motion on the plates of the capacitors. The capacitance deviation caused by the oscillating motion was monitored using a high frequency AC signal and an op-amp based capacitance sensing circuitry. The signal is optimized in a way that the capacitance vs. time waveforms are sensitive to changes in the mechanical properties of the accelerometer. From the capacitance measurements, the mechanical parameters of interest are predicted with less than 5% error using multivariate adaptive regression splines [44]. The method presented in their work and shown in Figure 6 is referred to as alternate testing [45].

While it is not a fully integrated BIST solution and requires external components, signal processing and evaluation off-chip, it is amenable to be one, as BIST solutions with the same test method were previously presented for RF ICs [46]. The test method, when fully on-chip, would alleviate the need for expensive measurement setup thus saving money on testing. Assuming that the mapping algorithm needs at least 1 full period to match the output to the physical properties of the device, the prediction of the device parameters should take around 22–25 ms. Although the technique was designed originally for accelerometers, it should be applicable to any capacitive MEMS.

### 2.4. Indirect Parameter Extraction Test

#### 2.4.1. Oscillation-Based Test

It is computationally easier and faster to use an oscillating signal to measure these electrical parameters that are highly correlated to the MEMS dynamics [30]. These include resonant frequency, pull-in voltage or amplitude response. This method, called the oscillation-based test method (OTM), was studied by Beroulle et al. at LIRMM [47,48] and Al-Gayem et al. at Lancaster University [49,50,51,52,53,54].

In the former case, OTM for analogue circuits was adapted to simulate, with polysilicon strain gauges, the deflection of a U-shaped cantilever vibrating due to Lorentz forces in the presence of a magnetic field, as shown in Figure 7. The magnetometer was reconfigured into an oscillating device using a first-order derivation feedback circuit to induce motion of the DUT and calculate the resulting frequency and the amplitude of the oscillations. Frequency and amplitude are referred to as indirect parameters, while the thicknesses of various layers of the cantilever structure are referred to as low-level parameters. Both the direct parameters such as sensitivity and settling time and the indirect parameters are dependent on the low-level physical parameters. OTM did not prove to be useful in reliable detection of small deviation of the low-level parameters, or small parametric faults, however it performed well for detecting catastrophic fault or large parametric faults. With a natural frequency of 11 kHz, the authors concluded that the duration of the test would be around 12 ms.

In the latter case [49], OTM was also used to simulate the degradation of the electrode surface in a microfluidic system by modelling the electrical double layer as a capacitance inserted into the feedback of a standard oscillating structure. The increase of frequency of the oscillator indicated the corrosion-induced reduction of the surface area, and therefore of the capacitance of the electrode. The experiments showed that 50% capacitance change resulted in a three-time increase of the oscillation frequency. The oscillation frequency for a fault-free electrode was above 20 kHz, thus assuming that at least one full cycle is needed for a measurement; the test time is under 50 µs.

Whereas both works were simulations, Al-Gayem et al. carried out physical experiments, albeit without an integrated solution. Neither OTM has been implemented for BIST yet, but has the potential to be a fully integrated BIST method [49].

#### 2.4.2. Impedance Analysis

Liu et al. [55] and later Al-Gayem et al. [49] presented another electrode degradation detection method based, this time, on the impedance analysis of a capacitive biosensor device. As the electrode degrades, the impedance increases and signal-to-noise ratio (SNR) drops. Their DUT had multiple sensing electrodes assumed to be fault-free. The impedances are measured and compared using a half-bridge structure [49]. The pass/fail decision is based on the ratio of the measured impedances and the reference. The ratiometric measurement demonstrates that the comparison method reveals damaged electrodes even if the reference electrodes are faulty, as illustrated in Figure 8 and Figure 9.

The impedance measurement was chosen to run at 1 kHz, but the theoretical assumptions for the impedances will hold true till the MHz range. Although the speed of the measurement system is currently in the ms range, tests durations of a few µs could be achievable.

### 2.5. Discussion

Test methods based on electrically induced stimuli alleviate the need for external physical stimuli. In order to implement the technique in a truly BIST manner, the test signal generation and response analysis must be implemented on chip [25]. With such methods, BIST could implement functional tests, focusing on measuring device specifications, and structural tests that focus on identifying defects. An example of the former is the periodic self-calibration of the Analog Devices accelerometer using electrostatic actuation and Earth’s gravity [26], as recommended by Zimmerman et al. [31]. A symmetrical test is a structural test method whose duration is device dependent, and which requires a modular MEMS structure that is specific to certain transducer types such as accelerometers or RF MEMS switches [56]. The redundancy offered by device symmetry may be utilized for BISR [57] applications if a suitable BIST method is present. The technique offers a better fault coverage when used in combination with electrically-induced stimuli test [42]. Direct parameter extraction method is a functional test allowing the direct extraction of the mechanical properties of the DUT using mapping techniques. Not yet implemented as a BIST method, the test duration depends on the selected frequencies, themselves dependent on the geometry and material properties of the device. In contrast, the indirect parameter extraction method is a structural test method which requires integrated evaluation circuitry to make is a truly BIST technique. In all of these methods, the dedicated BIST circuitry could be part of an ASIC heterogeneously integrated with the MEMS sensor in a package if necessary.

No “best BIST” solution exists for this category of tests, and the most suitable method needs to be selected on the user needs and limitations of the device. For example, a symmetrical test method is not suitable for non-modular devices. Direct parameter extraction might be able to extract vital information about the mechanical properties of the system at the expense of significantly more additional circuitry for signal processing and decision compared to other methods.

## 3. Delay-Based Test Methods

Delay-based test methods have been used extensively in microelectronics [58,59,60]. To the best of the authors’ knowledge, their application in capacitive MEMS BIST was originally carried out by Rashidzadeh et al., who utilised delay lines to measure time intervals which are a function of the capacitance of the DUT [61,62,63,64,65,66]. The test methods, described in this subsection, include the charge control method, phase-locked loop (PLL) method, phase locking method and the use of a Pierce oscillator.

### 3.1. Charge Control Technique

As shown in Figure 10, this BIST scheme utilizes a charge pump with two current sources that gradually charge (current source *I_1_*) and discharge (current source *I_2_*) the capacitance of a MEMS while measuring the charge and discharge times with delay lines [65]. A comb drive structure was used for testing, thus charging the device altered the electrostatic force between the plates. The movement of the plates results in a capacitance variation that is detected and converted to time intervals using delay lines. The time intervals are digitized by a time-to-digital converter (TDC) [67]. Some structural defects may alter the measured capacitance of the comb drive by changing its geometry and spring constant. These defects can include missing fingers, finger dimension dissimilarity and etch variation.

This method provides a safe BIST solution, without the risk of collapsing the suspended structure with electrostatic forces while being insensitive to noise and power supply variations. The demonstration chip, fabricated with the TSMC CMOS 65 nm process, occupied 0.7 mm^2^ silicon area and consumed less than 2.8 µW power for a maximum self-test speed of 18 kHz. The device was prototyped without a capacitive sensor but contained a CMOS varactor mimicking the behaviour or a MEMS capacitor. The chip is capable of measurements with ns resolution and can detect capacitance changes in the fF regime.

### 3.2. PLL Based Built-in Self-Test (BIST)

In this method, a phase-locked loop (PLL) transfers the capacitance variations of a capacitive MEMS in the time domain [64]. Inserting the MEMS between a phase-frequency detector (PFD) and a voltage-controlled oscillator (VCO) creates a phase difference between the input and output of the VCO, as shown in Figure 11. A change in the MEMS capacitance from its nominal value introduces a phase difference in the PFD inputs. This results in the PLL losing the lock, which it reacquires by VCO adjustment. In this state, the inputs of the PFD are in phase again, however the phase difference between the input voltage and the output of the VCO is different from its original value. This difference is linearly proportional to the deviation of the MEMS capacitance. The proposed method is insensitive to process, supply voltage and temperature (PVT) variations. Only simulations were carried out to validate the concept as a BIST method. This technique has measurement resolution in the ns regime and is capable of detecting changes in capacitance down to tens of aF.

### 3.3. Phase Locking Test Method

In this method, a phase-locking circuit is utilized to detect physical defects in a capacitive MEMS [63]. Small defects that change the capacitance by fFs are hard to detect, especially in the presence of PVT variations. To measure such small defects, a modified delay locked loop (DLL), chosen for its better jitter performance compared to a PLL, is used to amplify the capacitance variations in the time domain.

The parasitic capacitances of a typical MEMS actuator are in the fF range. This value is comparable to that of the delay cell capacitances that essentially define the resolution of a delay locked loop. As the DLL resolution is in the parasitic capacitance range, smaller deviations to the MEMS capacitance caused by minor defects may go undetected. To increase its resolution, the DLL was reconfigured into a delay detection module (DDM) so that the resolution of the circuit now depends on the delay difference between two delay cells and not on the delay of each cell. This allows one to amplify the delay introduced by the MEMS capacitor, as shown in Figure 12.

A prototype circuit of an estimated area of 7600 μm^2^ was realised in 0.18 µm CMOS technology for performance verification. This technique operates in the ns time regime and is capable of detecting up to 2 fF capacitance variations.

### 3.4. Pierce Oscillator

Unlike other delay-based test methods, this BIST method, used for capacitive MEMS, operates in the frequency domain instead of the time domain [61]. In [58], structural defects causing very small capacitance deviations are claimed to be easier to detect in the frequency domain than in the time domain, especially for resonant structures monitored closed to their resonant frequency. In that regard, a Pierce oscillator structure is utilized to assess capacitance deviations in the frequency domain.

In the study of Rashidzadeh et al., the resonance frequency of the DUT depends on the MEMS capacitance. As shown in Figure 13, a 10 fF change in a capacitance of a comb drive structure with a nominal value of 1 pF creates a 10 MHz deviation of a resonant frequency of 1 GHz, which is much easier to detect than the corresponding deviation in the time domain. This BIST duration is in the order of ns and the method is able to detect structural failures with high accuracy, such as missing fingers in a comb-drive.

### 3.5. Discussion

All delay-based tests methods highlighted above were applied for comb drive structures but could equally be used for other capacitive MEMS. The methods utilise delay lines to measure time delays associated with MEMS capacitances and operate within the ns duration. They convert capacitance variations into time differences. The Pierce oscillator translates these capacitance variations into the frequency domain, while the phase locking test method amplifies the time difference using a “time amplifier”, which is a combination of a delay detection module and delay lines. It could be argued that the PLL-based BIST is not strictly a delay-based method but this test shows great similarity to the phase locking test solution. The phase locking method has however better jitter performance. Apart from the Pierce oscillator method, all are insensitive to PVT variations. As the experimental validation of methods were carried out using different CMOS processes, power consumption and die overhead are not directly comparable. The PLL-based BIST method is superior in detecting small capacitance deviations as it proposes aF level resolution, while the rest of the methods only offer fF resolution. Apart from the PLL-based BIST solution, all delay-based test methods were fabricated and experimentally validated. Although few BIST solutions exist for RF MEMS [68,69,70], it was suggested by Syed et al. [71] that non-contact capacitance deviation detection methods can be utilized to detect certain failures such as stiction, thus delay-based test methods may be suitable candidates for RF MEMS BIST.

## 4. Impulse Response Test Method

One of the simplest ways to assess the health of a linear time invariant (LTI) system is to calculate its impulse response (IR). The IR of a system can be acquired by applying a Dirac (delta) δ-function as the system input and measuring the output response. By knowing the IR of the system, the output for any arbitrary input can be calculated, thus failure-related deviations can be identified. Various strategies were implemented based on this concept; these include the pulse excitation test and the pseudorandom test methods.

### 4.1. Pulse Excitation Test Method

An approximation of the δ-function is a pulse signal. This method enables fast system characterization, but has a very low signal-to-noise ratio due to its short time duration [72]. Rarely used since more accurate, albeit slower, methods exist to calculate the IR of a system, pulse excitation is however a feasible solution for time-critical applications.

A patent from InvenSense Inc. describes an on-chip pulse excitation (PE) solution for testing a capacitive microphone as shown in Figure 14 [73]. The electrical pulse is generated by an electrostatic force generator. The circuit includes a voltage buffer to drive the test signal onto the output of a high voltage charge pump (HVCP). The output of the voltage buffer is coupled through a capacitor so that it does not disturb the DC voltage at the output of the HVCP. A resistor is used to control the time constant of the applied waveform.

### 4.2. Pseudorandom Test Method

A more accurate, higher SNR but slower test method uses a pseudo-randomly coded test sequence to calculate the impulse response of a system. The impulse response is obtained from the measurements of the input-output cross-correlation function [25,72,74,75,76,77,78,79,80].

One of the earliest pseudo-random testing techniques for BIST presented by Pan et al. [81] in Figure 15 uses a linear-feedback shift register to generate a maximum length sequence (MLS), which is the test sequence. MLSs are binary sequences with a length N = 2 ^m^-1, where m is the order of the sequence. This sequence is highly suitable for IR measurements, since it is periodic and deterministic. Only one period of the MLS signal is enough to calculate the impulse response in case of a linear time-invariant system. Averaging can be applied to increase SNR. Due to the deterministic property of the MLS, it is precisely repeatable. The SNR can thus be further increased with the synchronous averaging of the test output.

### 4.3. Discussion

It is hard to compare the methods due to the low number of available papers on pulse excitation. In general, PE is faster than pseudorandom testing and the injection of the test signal requires less additional circuitry. PE has, however, a lower SNR and there is a chance of damaging the MEMS device. Pseudorandom testing is more accurate at the expense of more additional circuitry and processing time. Mir et al. were able to calculate the impulse response of a bulk micromachined CMOS cantilever beam with this method under 1 ms [25].

## 5. Bias Superposition/Modulation Method

The electrically induced stimuli test methods presented in Section 2.2 are off-line methods since they perturb MEMS outputs during normal operating conditions [82]. A solution is required to test MEMS during their operation, whilst making sure that test results can be solely related to the input stimulus response. As shown in Figure 16, in the bias superposition method, also called bias modulation method, a test stimulus is either added (superposition) or multiplied (modulation) to the normal electrical biasing of the structure of the MEMS to be tested. The characteristic of the stimulus allows the test response to be extracted from the normal functional response of the MEMS, ideally without compromising either signal. The injected test signal has typically a frequency outside the operating bandwidth of the functional signal, but still within the physical bandwidth of the MEMS to allow online operation. Following [82], this subsection describes in detail the four different types of stimuli used for these tests: differential bias, alternative bias, actuating bias and redundancy bias.

### 5.1. Differential Bias

The differential bias method relies on device symmetry to identify structural defects. In the case of a pressure sensor, a Wheatstone bridge structure, Figure 17, is used with piezo resistive strain gauges experiencing increase and decrease of resistance depending on the applied strain [82]. The voltage supply of the structure is split into two differential inputs containing the bias voltage and an ac signal of different polarity but same magnitude. The AC output is independent of the strain applied to the structure as the values of the resistors stay relative to each other along with the rate of change. If the change only affects a single resistor, the AC output will change, and the fault is detected.

### 5.2. Alternative Bias

The alternative bias method utilizes a signal superimposed or modulated to the bias signal that excites the sensor element in a way distinct from its normal mode of operation. This method exposes defects without any measurement. In the case of a MEMS magnetometer based on a U-shaped free-standing mechanical frame connected to a Wheatstone bridge as in Figure 18 [85,86], a low-frequency signal is injected for thermal excitation purposes. Lorentz forces actuate the structure at its mechanical resonance frequency of around 22.5 kHz. A low frequency signal of 4.5 Hz injected to the bias current (or voltage) induces thermal variations to the mechanical resonance. Both components can be read at the output of the sensor. In case of a catastrophic failure, the thermal dissipation of the bridge structure is likely to change; so does the low frequency output. This method could also be included in Section 2.1.2.

Both bias superposition and modulation strategies have been investigated. For bias superposition, the amplitude increase of the output is caused by asymmetrical bias. For bias modulation, the output of the sensor is proportional to the peak-to-peak amplitude of the bias signal. Assuming that at least 1 full cycle is needed for measurement evaluation, the duration of this test method is around 250 ms.

### 5.3. Actuating Bias

The actuating bias method utilizes a signal generated by the bias to physically excite the sensor in the same manner as the bias itself. Either the bias signal generates a similar response to the measurand (same type of output/effect) or mimics the response of the measurand (same type and magnitude). Simulation work was carried out in the case a capacitive MEMS accelerometer with the test stimulus lying in the mechanical bandwidth of the DUT, as shown in Figure 19 [82]. The superimposed signal must not, however, influence the measurement output, and must be separable by filters. Experiments were later carried out by Dumas et al. with the frequency of the injected stimulus sitting in the mechanical bandwidth of the device, but outside the bandwidth of interest [88]. Since the measurand-induced signal and the test signal have different frequencies, they are simply separated by filters at the output of the DUT. This case study also highlights the narrow bandwidth of the test stimulus and the requirements for high roll off test filters.

Instead of employing a sinusoidal signal for the injected stimulus, improvement on the readability of the output was demonstrated by using a pseudo-randomly coded sinusoidal wave generated by a linear feedback shift register (LFSR) [81]. Two scalar values, covariance (Cov) and correlation (Cor), were produced at the test output and are generated by covariance and correlation algorithms, respectively. Cov is proportional to the sensor sensitivity, while Cor is used to validate the value of Cov and serves as an indicator for output signal corruption. The health of the system is determined by the value of Cov, which indicates whether system sensitivity falls below a threshold value. The pseudorandom coding ensures that Cov is readable on the output. This requires a trade-off between readability and test speed as increasing the number of bits used for the pseudorandom code improves the readability of the signal. Dumas et al. found the number of minimum acceptable bits for their case study to be 255, resulting in a test duration of 2.55 s.

### 5.4. Redundancy Bias

In the redundancy bias method, the bias signal is manipulated to support two distinct measurements of the physical input. The two measurements are correlated to obtain data that constitute a function of the structural MEMS parameters.

Application of this method was carried out on a thick film conductivity sensor used in electrolyte conductivity measurements [89]. The sensor uses two gold electrodes around a micro-channel, with the channel sidewalls exposed to the electrodes as shown in Figure 20. To measure the resistance of the solution in the channel, high measurement frequency (10 kHz) is desirable to avoid electrolysis and the unwelcome impedances of the contacts at the electrode-solution interface and at the grain boundaries due to the unideal flatness of the electrode surface. In order to measure the time constant of the system, a 10 Hz step signal is required that activates the transient response. Jeffrey et al. demonstrated a relationship between the measured time constant and the concentration of the solution. By measuring the time constant, information about the concentration can be acquired, that is used to calculate a second value for the solution resistance. The high frequency (10 kHz) and low frequency (10 Hz) signals are mixed together and used as sensor bias, as illustrated in Figure 21. Measurement of the solution resistance (R’_x_) is achieved using the high frequency signal and the time constant is measured using the low frequency signal, from which a second value for the solution resistance is calculated (R’’_x_). A mismatch between R’_x_ and R’’_x_ or a time constant outside the specifications stored during sensor calibration indicates DUT failure. This provides pass/fail information about the health of the sensor.

### 5.5. Discussion

In summary, bias methods have the advantage of not requiring modification of the MEMS sensor to test. These online test methods also allow failure detection during online MEMS operation. In this respect, test durations ranging from 2.5 s (actuating tested using pseudorandom code sinusoidal signal) down to sub-ms (differential bias) become less critical, as the tests do not disturb the normal use of the MEMS. Extraction of the output signal resulting from the input test signal is feasible with electrical filters as the former injected signal has a frequency outside the operational frequency of the MEMS. This output signal provides information on the health of the MEMS and potential online signal compensation in case of calibration error.

As only pass/fail information can be extracted, these test methods are not suitable for MEMS characterisation or parametric measurements, although they provide a cost-efficient low-overhead solution for catastrophic fault detection. Regarding the use cases, differential bias can only be implemented when a symmetric/modular structure is present, and for failures that do not affect MEMS in a symmetrical manner such as stiction. As in the symmetrical test method (Section 2.2), it is best used with another test method. Alternative bias and redundancy bias are both easy to implement, however the readout circuitry of redundancy bias might be a bit more complex. Actuating bias is the slowest option and a working solution requires not only a test signal evaluator, but an additional test signal generator circuitry on board as well.

## 6. BIST Method Building on Multi-Functional Sensors

Redundancy of components in an electronic device leads to fault tolerance. If, after the implementation of a BIST method, a defective component is detected, one strategy is to isolate this faulty component from the rest of the device and carry on device operation using the remaining functional components [90].

In the case of MEMS, the manufacture of multiple identical sensors on the same die might be space or cost prohibitive. However, if the DUT has already multiple sensors for the purpose of sensing more than one physical parameters, one strategy is to use the inherent redundancy of sensing modalities possessed by these sensors that are sensitive to physical quantities purposely measured by other sensors. The effect when sensing elements are sensitive to more than one physical quantity is called cross-sensitivity, while the sensors themselves are called multi-modal sensing elements [91]. Cross-sensitivity is usually considered a problem in multi-sensor structures and approaches are made to overcome its effects [92]. For BIST, cross-sensitivity can be exploited as a substitute for redundancy. Extensive research has been carried out in the field of multi-functional sensors (MFS) at Lancaster University by Richardson et al., an example of which is provided in Figure 22 [93,94,95].

In this figure, each sensor has not only a primary sensing function but can also measure secondary parameters through either additional modes or cross-sensitivities. For example, the sensor on the left would sense primarily pressure, but also temperature. The sensor in the middle would sense temperature but also humidity, and the sensor on the right-hand side would sense humidity but also pressure. The use of multiple MFS is to increase fault tolerance, such that, if one sensor fails, the system tries to recreate the correct measurement of humidity, pressure and temperature by using the outputs of the remaining sensors. In order to achieve this, the measurement data from each sensor are first pre-processed and then passed through a data fusion algorithm to compare the measurements from each sensing element and produce an output. Pre-defined coefficients are used for sensor calibration at default. The data fusion algorithm detects whether a sensor is defective or not based on comparison.

This BIST technique can also be enlarged for built-in self-repair (BISR) in the following manner. When a faulty sensor is detected, sensitivity coefficients are calibrated for that sensor. If the modified coefficients lead to a sensor output that is in the acceptable range, the sensor output is used for further measurements (recovered from failure state); otherwise, the sensor needs to be replaced in the next maintenance. The data from the sensor coefficient recalibration algorithm can not only be utilized for monitoring the health state of each sensor, but also for prognosis, thereby enabling the calculation of the residual useful life (RUL) and end of life (EOL) of the MEMS [96,97,98].

In summary, this BIST method relies on the plurality of sensing modalities inherent in some sensors. The test method has the advantage of being extendable to MEMS BISR and prognosis. This method can, however, only be applied to MEMS capable of accommodating sensors with cross-sensitivity with different sensing elements.

## 7. Classification and Benchmarking of the BIST Methods

Several performance metrics can be used to benchmark the various BIST methods presented in the previous sections. Some metrics such as test duration or power consumption of the test methods, are measurable quantities, although their values can be dependent on the technology used and SNR requested. Other metrics, like “ease of implementation” or “usefulness”, are more subjective. Such metrics can nevertheless be provided with an ordinal value set up by consulting engineering teams. In the case of “ease of implementation”, the range was from 3 (easy) to 1 (difficult). A 5-point cumulative score was used for the “usefulness” metric, depending on the type of faults covered by the BIST method and the additional functionalities that the method can provide potentially. A high score for each of these metrics indicate therefore that the method is highly desirable. These last two metrics are first described and presented in Table 1 and Table 2, followed by the measurable performance metrics.

Following this benchmarking exercise, the various applications of the BIST methods are presented and classified according as to whether they are run during system operation, at the device assembly phase or during power-on. Finally, a classification table summarising the findings of this review article is provided in Table 3 at the end of this section.

### 7.1. Benchmarking Methodology of the BIST Methods

This subsection introduces three benchmarking criteria for BIST methods. The criteria were designed to provide a quantitative evaluation between the different methods that are otherwise hard to compare.

#### 7.1.1. Ease of Implementation

The ease of implementation of a test method for MEMS is an important consideration for selection of the BIST techniques. Some methods require either slight modification or radical redesign of the MEMS when additional sensors are required, resulting in additional real estate on the die. Other methods require new test signals or modification of existing signals necessitating perhaps additional electronic circuitry. A quantification of this “ease of implementation” is presented here and consists of a scoring scheme with each level assigned to a level of difficulty, ranging from 1 (difficult) to 3 (easy). The scoring method is presented in Table 1 and enables the objective scoring of this performance metric.

#### 7.1.2. Usefulness

The degree of “usefulness” of a BIST method for MEMS was broken into two main categories: whether it supports parametric fault detection or catastrophic fault detection only. The latter property provides only a pass/fail information about the MEMS, whereas the former can be used to detect deviations from system specifications. Parametric fault detection methods are the pillars for embedded intelligence in MEMS. These techniques make it possible to include additional support and evaluation tools, such as on-line health monitoring, BISR, RUL, self-tuning and EOL prediction. Multi-modal sensing was included in the evaluation list due to its benefits in not only in BIST but BISR support. The two main categories each are worth 1 point and one point is awarded for each additional support tool included. This creates a ranking system from 1–5 with 1 being the lowest score and 5 being the highest. The scoring system is shown in Table 2.

#### 7.1.3. Test Duration

Test duration defines the domain of application of the BIST method with tests with long duration are preferably applied when the device is powered on. This quantity is less important however in cases where the test method does not perturb the normal operation of the MEMS and is running concomitantly. The durations of the test indicated in the classification table have been either taken from literature or estimated.

#### 7.1.4. Power Consumption

Power consumption is a critical aspect of the test. The MEMS should not have heat dissipation problems due to increased power demand from the test. Depending on the application and assuming that the BIST method is applied to mobile devices, the test should not put an undue increased load on the battery. Unfortunately, no data were available regarding power consumption for the various BIST methods reviewed. The classification table therefore does not figure this important performance metric.

### 7.2. Classification of the BIST Methods

The applications of the BIST methods have been categorised as to whether they are implemented on the run (field test), during power-on of the system or during the manufacturing/assembly phase of the MEMS.

#### 7.2.1. Field Test

Tests happening on the run are referred to as field tests. The device is in use or assumed that it can be used anytime. The test duration must therefore be very short as the test must not interfere with the user experience of the device. A typical test duration of less than 100 ms would be deemed to be acceptable. Test methods used in field tests have been labelled A in the classification table.

#### 7.2.2. Power-On Test

A power-on test occurs during the switch-on phase of the system. As such, an extended amount of test time is available and could range from 100 ms up to 30 s. Test methods in that category have been labelled B in the classification table.

#### 7.2.3. Assembly Phase Test

Testing happening during the manufacturing or assembly phase of the device has virtually unlimited test time and power consumption. The device is not in use and is usually connected to an external power source. A label C has been indicated for these types of test methods in the classification table.

## 8. General Conclusions

From this review, it is quite clear that there is no absolute “best BIST” method. The most suitable test technique for implementation is likely to be based on the MEMS sensing and actuating properties and the domain of application, as well as the MEMS failure modes to be monitored or detected. Some of the BIST methods come with an increased number of manufacturing steps and with an overhead in term of die area, resulting perhaps in cost increase. This review article did not attempt to quantify such costs. The expenditure introduced by the additional functionalities should be offset however by the reduced yield loss and increase device lifetime provided by the BIST methods. A quick summary of the advantages of the various methods is provided in this final section and should be read alongside the classification table.

Parameter extraction comes with probably the most complex circuitry regarding evaluation of the output signal. However, the test method can extract important information regarding physical parameters of the device and enable RUL estimation. Symmetrical test method is only advised in combination with single-ended testing and is able not only to detect hard to detect faults, but to characterise/configure the system provided that it has some form of symmetry. For certain systems, where the number of failure modes is limited (e.g., electrode degradation), indirect parameter measurement can be a good an inexpensive option. Delay based methods present one of the fastest ways of self-testing. Only suitable for capacitive MEMS and therefore inadequate for complete system characterisation, they require additional circuitry to convert capacitance variations into time differences. It is hard to draw conclusions about pulse excitation due to the lack of literature on MEMS applications, but the pseudo-random test method shows great promise on system impulse response extraction, thus it is suitable not only for catastrophic fault detection, but also for system characterisation. The test time duration depends on the number of bits used for the MLS, which is device-dependent.

All the previously mentioned methods are intrusive and require a normal operation of the device to stop while the test is running. A great alternative is the superposition test methods that offer an on-line test solution mostly with minimum overhead. They are capable of catastrophic fault detection. Finally, multimodality sensing, albeit not a requirement for BIST, can play an important role in BIST and BISR. Through cross-sensitivity, the overhead of redundancy can be reduced while maintaining its benefits.

This review article approached BIST strategies from a single device point of view, however one might not be limited to a particular MEMS unit. The BIST method could easily take place in a different system component, along with the electronic processing enabling decision making. With proper communication between the components of the system, the evaluation and decision making could easily run on a processor that is dedicated for another work, during its idle period. Such communication approaches and the required interfaces already exist. One of these approaches is coming from the MIPI alliance, an organisation working on hardware and software interfaces that make the integration of components into a mobile device simpler [99]. Perhaps such an approach will pave the way for the future of BIST methods in MEMS.

## Figures and Tables

**Figure 1 micromachines-12-00040-f001:**
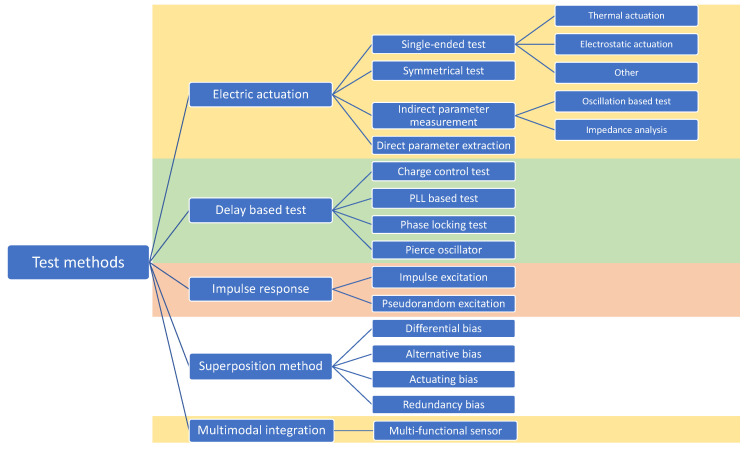
Taxonomy of the built-in self-test (BIST) methods for micro-electro-mechanical systems (MEMS).

**Figure 2 micromachines-12-00040-f002:**
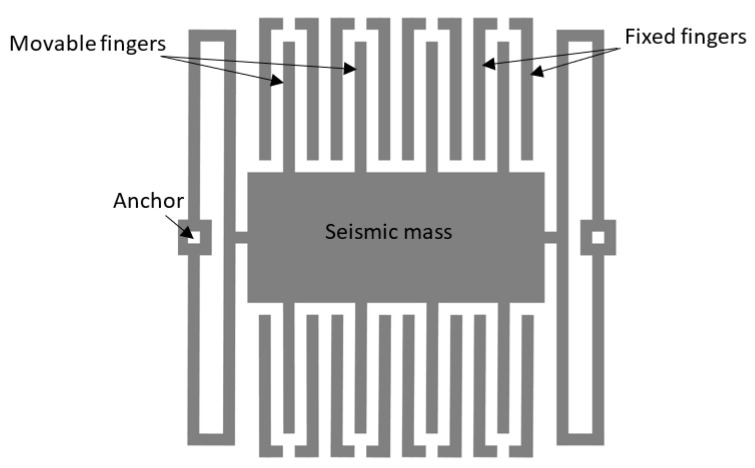
Schematic of a capacitive accelerometer, adapted from [23].

**Figure 3 micromachines-12-00040-f003:**
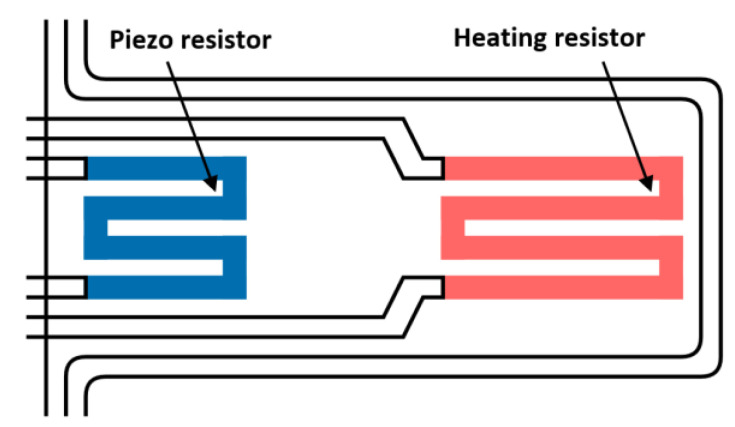
Schematic of a meander heater electrode in a cantilever beam, adapted from [23].

**Figure 4 micromachines-12-00040-f004:**
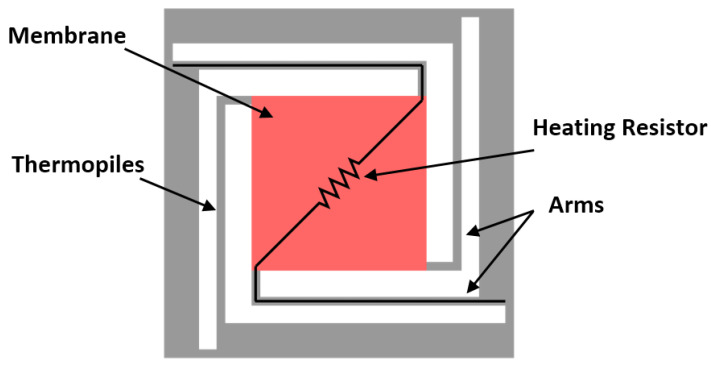
Schematic of a thermopile-based sensor with a heating resistor, adapted from [24].

**Figure 5 micromachines-12-00040-f005:**
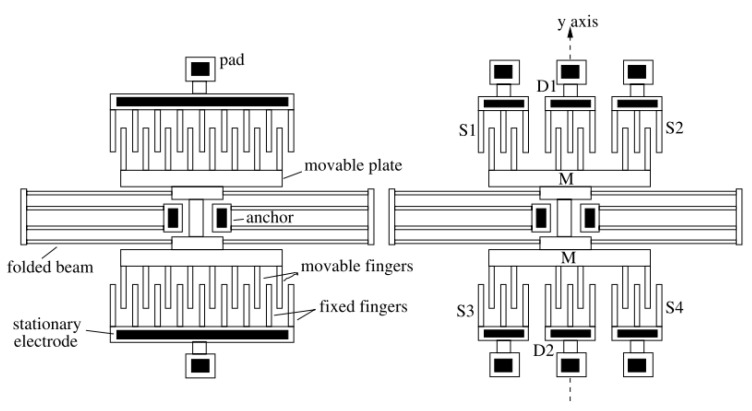
MEMS comb drive structure before (**left**) and after (**right**) partitioning [41].

**Figure 6 micromachines-12-00040-f006:**
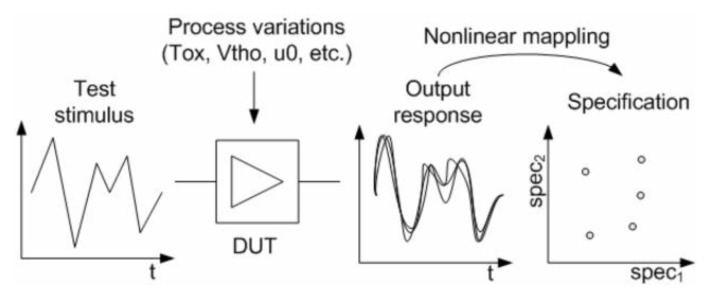
Alternate test methodology using output response mapping [46].

**Figure 7 micromachines-12-00040-f007:**
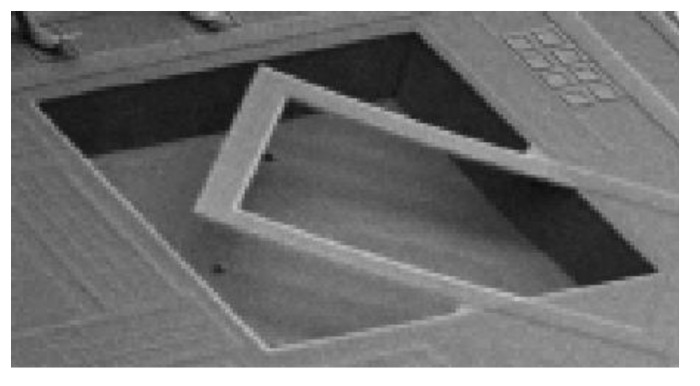
Cantilever-based magnetometer structure [48].

**Figure 8 micromachines-12-00040-f008:**
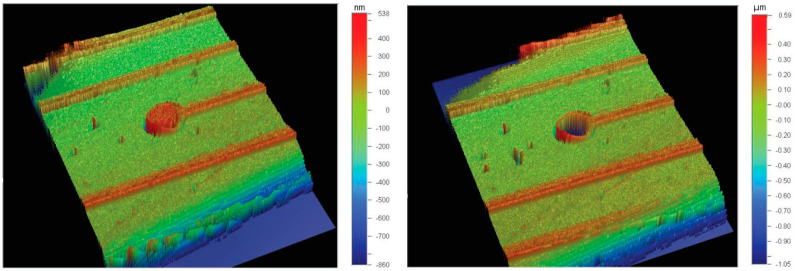
Interferometry images of fault-free (**left**) and degraded (**right**) electrodes [50].

**Figure 9 micromachines-12-00040-f009:**
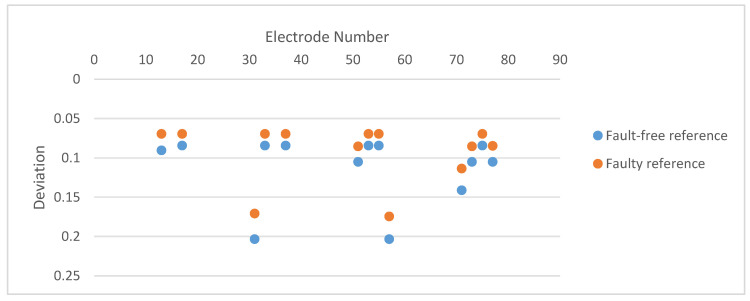
Deviations of impedance ratios. The damaged electrodes are no. 31 and no. 57. Adapted from [52].

**Figure 10 micromachines-12-00040-f010:**
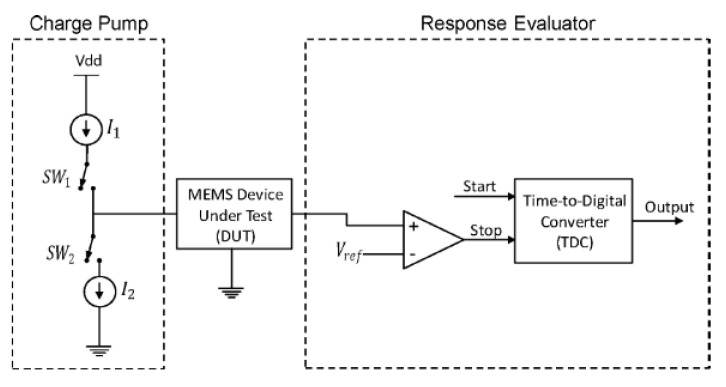
Simplified block diagram of the BIST architecture [65].

**Figure 11 micromachines-12-00040-f011:**
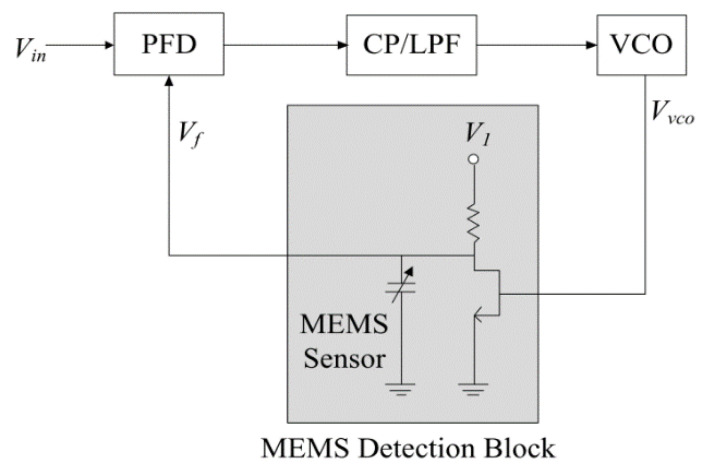
Block diagram of the PLL-based readout circuit [64].

**Figure 12 micromachines-12-00040-f012:**
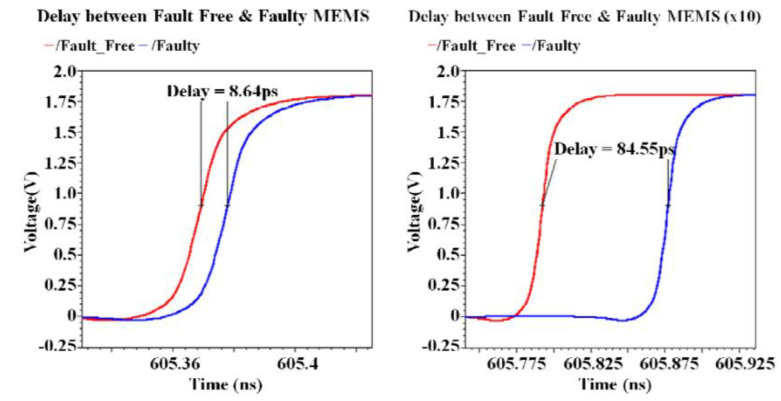
Output response difference between fault-free (**left**) and faulty (**right**) device [63].

**Figure 13 micromachines-12-00040-f013:**
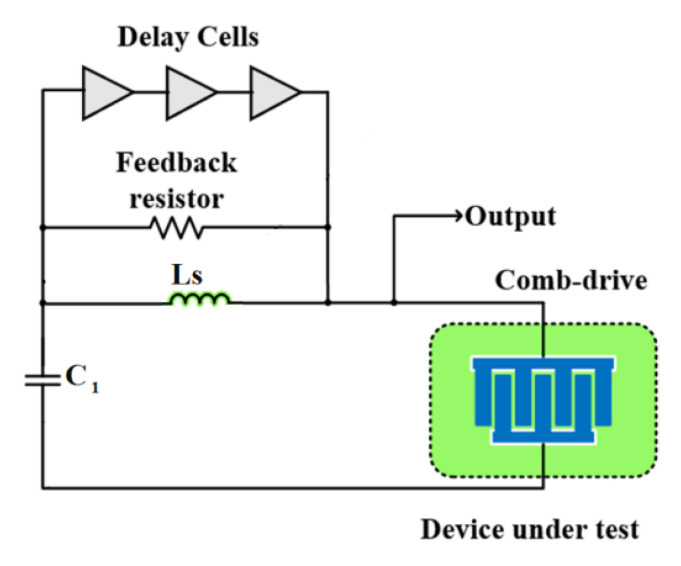
Block diagram of the proposed BIST solution [61].

**Figure 14 micromachines-12-00040-f014:**
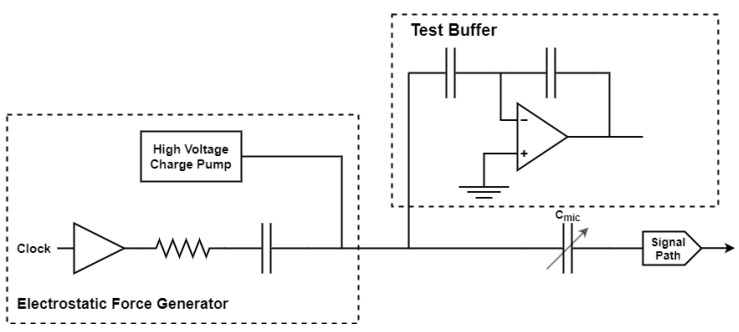
Schematic of pulse excitation-based test solution for a capacitive microphone, adapted from [73].

**Figure 15 micromachines-12-00040-f015:**
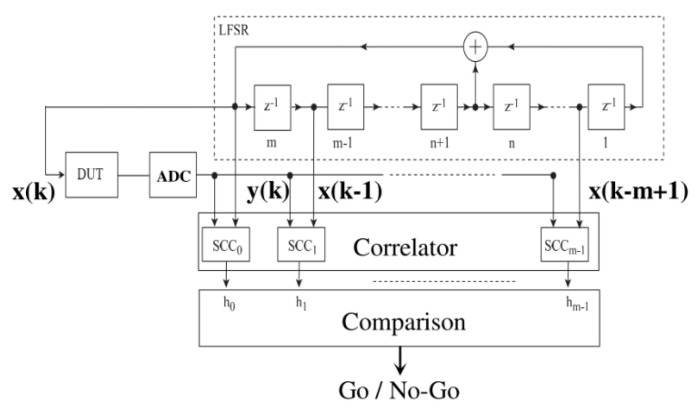
Block diagram of pseudo-random test approach [80].

**Figure 16 micromachines-12-00040-f016:**
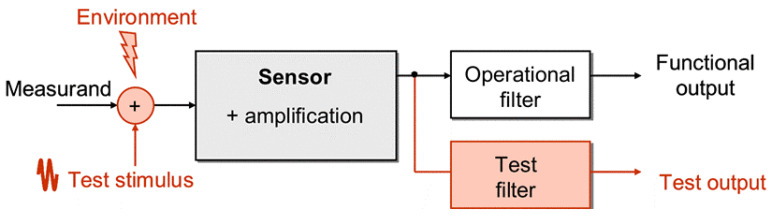
Test scheme for bias superposition/modulation [83].

**Figure 17 micromachines-12-00040-f017:**
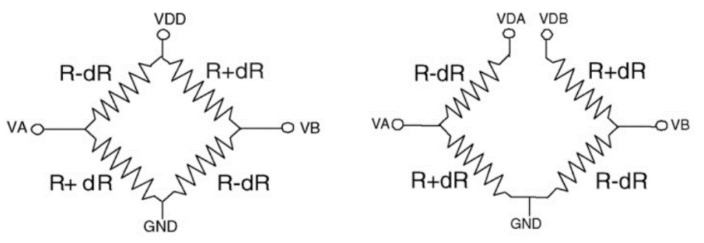
Wheatstone bridge with normal (**left**) and differential (**right**) input [84].

**Figure 18 micromachines-12-00040-f018:**
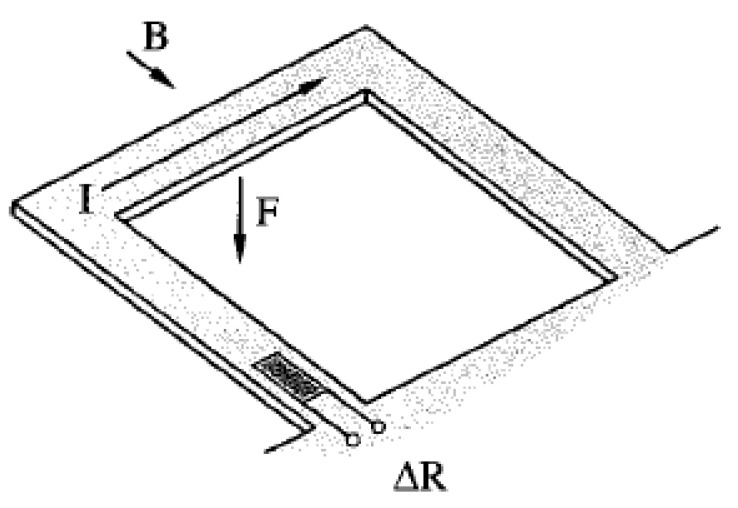
Schematic view of the sensor structure [87].

**Figure 19 micromachines-12-00040-f019:**
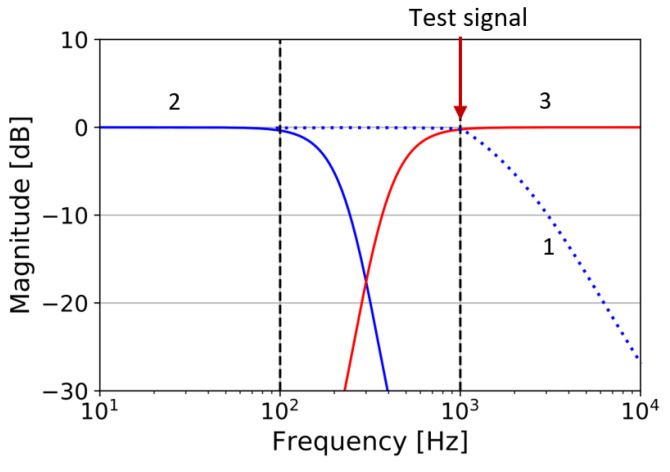
Normalised frequency responses of the sensor (1), the operational filter/low pass filter (2), the test filter/high pass filter (3) [83].

**Figure 20 micromachines-12-00040-f020:**
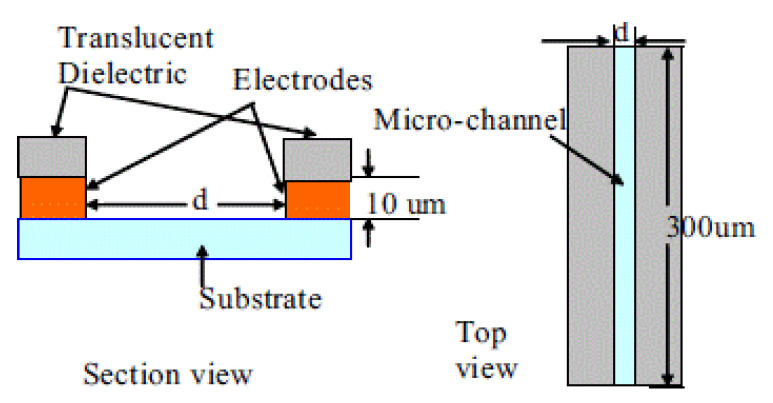
Conductance sensor on alumina substrate [84].

**Figure 21 micromachines-12-00040-f021:**
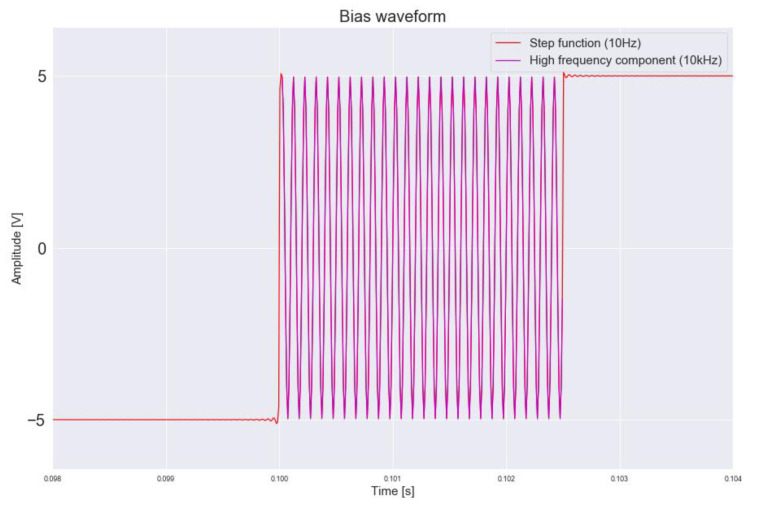
Redundancy bias waveform, adapted from [84].

**Figure 22 micromachines-12-00040-f022:**
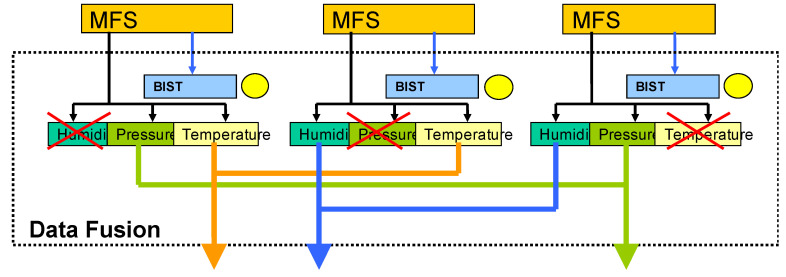
Schematic of a Multi-Functional Sensor (MFS) system. Traditional BIST methods (blue box) can be implemented at the level of each MFS using some of the techniques described in previous sections [93].

**Table 1 micromachines-12-00040-t001:** Quantitative scoring method for implementation difficulty of built-in self-test (BIST) methods.

Difficulty	Description	Scoring
Easy	No MEMS alteration is necessary; no sensing elements are needed to be manufactured and only additional electronic circuitry is needed if necessary.	3
Medium	Alteration of the MEMS structure is minimal. Additional circuitry may be required.	2
Difficult	Additional sensing elements are needed and/or major modifications of the MEMS structure are required.	1

**Table 2 micromachines-12-00040-t002:** Quantitative scoring method of BIST usefulness.

DUT Support Type	DUT Support Function	Scoring
Base fault detection	Catastrophic fault detection	+1
	Parametric fault detection	+1
Additional support tools	Health and usage monitoring	+1
	Built-in self-repair capability	+1
	Multi-modality sensing	+1

**Table 3 micromachines-12-00040-t003:** Classification table BIST methods.

Test Method			Description	Ease of Implementation	Usefulness	Measurement Time [ms]	Application of Test	References
**Electric actuation**	**Single-ended test**	**Thermal actuation**	Heater electrode is used to provoke thermally induced deformation.	2	2	20,000	B	[23,24,27,28]
		**Other**	Heater electrode is utilized to generate pressure gradient via heating air that causes membrane deflection or to provide signal to infrared imager.	2	2	70, 7000	A, B	[23,24,29,36]
		**Electrostatic actuation**	Electric stimulus is used to mimic physical force, in most cases acceleration.	2	2	10	A	[23,24,26,30,31]
	**Symmetrical test**		Combination of single-ended and symmetrical testing for greater fault coverage.	2	2	<1	A	[41,42]
			Device is structured into symmetrical portions. Response is captured and compared for identical stimuli.	2	2	2	A	[37,38,39,40]
	**Indirect parameter measurement**	**Oscillation based test**	Device is reconfigured into an oscillating device, changes in resonating frequency and amplitude are captured.	2	2	15	A	[47,48,86]
			Electrode surface area change due to degradation results in capacitance drop that modifies the oscillation frequency of the system.	3	2	1.5	A	[49,50,51,52,53,54,55]
		**Impedance analysis**	Electrode surface area change due to degradation results in impedance deviation.	3	3	0.1	A	
	**Direct parameter extraction**		Electrode impedances compared to the reference electrodes to monitor degradation.	3	2	0.1–10	A	[43]
**Delay based**	**Charge control test**		Charging and discharging the MEMS capacitor, test is based on charge times.	3	2	1–10	A	[65]
	**PLL based test**		Change in MEMS capacitance detected by a PLL and converted to time domain.	3	2	1	A	[64]
	**Phase locking test**		Amplification of capacitance differences in time domain using a modified DLL.	3s	2	0.007	A	[63]
	**Pierce oscillator**		Analysing capacitance deviation in the frequency domain utilizing a modified Pierce oscillator structure.	3	2	0.01	A	[61]
**Impulse Response**	**Impulse Excitation**		MEMS is excited with a single impulse. Impulse response (IR) gives information for functional evaluation.	3	2	2.5	A	[72,73]
	**Pseudorandom**		MEMS is excited with a test sequence to acquire IR of the system with better SNR than with IE.	3	2	2000	B	[72]
**Superposition method**	**Differential bias**		Relies on device symmetry. Separating bias signal to two different inputs. The output should be the same unless there is a change in resistance.	3	3	20–50	A	[82]
	**Alternative bias**		A signal superposed to the bias signal that excites the sensor element in a distinct way than the normal mode of operation.	3	3	20-50	A	[82]
	**Actuating bias**		A signal generated by the bias that excites the sensor element in the normal mode of operation.	3	3	20–50	A	[82]
			The bias is modulated with a pseudorandom code-modulated sin-wave in the working range of the device.	3	3	2550	B	[88]
	**Redundancy bias**		The bias signal is manipulated to support two distinct measurements of the physical input. The information acquired from the output is redundant and is correlated to gain data about the structural sensor parameters.	3	3	30	A	[82,89]
Multimodal integration	**MFS**		Multiple sensors on one chip with cross sensitivity result in fault tolerance. Data fusion algorithm detects and attempts to correct misbehaving sensors. Integrated remaining useful lifetime prediction. Proposal for integration of sensitivity test method and pseudorandom coding.	1	N/A	N/A	N/A	[93,94,95]

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
