# Peer review of "Built-In Self-Test (BIST) Methods for MEMS: A Review"

_micromachines, 2020, doi:10.3390/mi12010040_

Round 1

Reviewer 1 Report

This is a review paper which is presented with the intention to rise the concern of the essential need of self test strategies in MEMS as compact and complex structure which operate driven by multiple physical phenomena. The technical innovation of the paper is as new as the presented solutions are.

The subject of the paper is of great interest to designers of MEMS, as much as packaging is. The subject is not very much entertained in the literature as the subject is quite vast and it requires good understanding of multi-physics phenomena. I will be glad to see such a paper published in Micromachines.

What I noticed there are few kinds of designs which were not mentioned in the paper – the micro-accelerometer of the IC-MEMS – to mention just an example. For sure that s review must not discuss al the possible subjects. However, conceptually, that accelerometer represents a great design.

I also have a couple of suggestions to improve the quality of the manuscript: the outmost of importance is to introduction of a list of abbreviations after the introduction or at the end as the paper includes many of those.

Few direct suggestions:

Please, reformulate 301-309

Please reformulate 339-342

Please improve the quality of fig 19 if possible

Line557 – usually a title is not followed by another title – some text should be included between the two titles

In my opinion, the ranking of quite subjective aspects defined as “ease to implement” or “usefulness” cannot be compared in any way with crisp variables as minimum duration of test or power consumption. Probably, a qualitative evaluation could be included in the performance evaluation using numeric parameters. I see as another possible way to indicate ranges that are called as very performant, performant and less performant and bring all to a common “adjective” based performance evaluation.

Reviewer 2 Report

Authors have presented a review article for MEMS BIST methods. The article is well organized and reviewed, there are few details which authors should consider to make this article stronger in contrast with the currently available literature.

1. Authors should first define MEMS (sensors, RF-MEMS, actuators etc) and discuss briefly the various challenges (from a general point of view) of testing in comparison with solid-state devices. In the current manuscript, authors are starting this article by introducing BIST only.

2. Authors have not discussed RF-MEMS devices. RF-MEMS switches and components has a niche market and the companies are still in development stage to commercially sale RF-MEMS switches. See devices from Analog Devices Inc., and Menlo Micro (a GE division), Cavendish Kinetics, and WiSpry.

3. There are various complex devices published by researchers such as G. Rebeiz (UCSD), R. Mansour (Univ. of Waterloo), P. Blondy (Univ. of Limoges), G. Fedder (CMU) to name a few. 

Author should consider these recent works for inclusion in the revised article

a). T. Singh, N. K. Khaira and R. R. Mansour, "Thermally Actuated SOI RF MEMS-Based Fully Integrated Passive Reflective-Type Analog Phase Shifter for mmWave Applications," in IEEE Transactions on Microwave Theory and Techniques, doi: 10.1109/TMTT.2020.3018141.

b). T. Lin, K. K. Wei Low, R. Gaddi and G. M. Rebeiz, "High-Linearity 5.3-7.0 GHz 3-Pole Tunable Bandpass Filter Using Commercial RF MEMS Capacitors," 2018 48th European Microwave Conference (EuMC), Madrid, 2018, pp. 555-558, doi: 10.23919/EuMC.2018.8541669.

c). J. -C. Orlianges, M. Laouini, C. Hallepee and P. Blondy, "RF-MEMS Switched Capacitor using Ta/Ta2O5 Electrodes," 2020 IEEE/MTT-S International Microwave Symposium (IMS), Los Angeles, CA, USA, 2020, pp. 41-44, doi: 10.1109/IMS30576.2020.9223836.

d). Y. Lin, M. G. Guney and G. K. Fedder, "ALD Titania Sidewalls on a CMOS-MEMS Resonator Oscillator and Effects on Resonant Frequency Drift," 2019 IEEE 32nd International Conference on Micro Electro Mechanical Systems (MEMS), Seoul, Korea (South), 2019, pp. 640-643, doi: 10.1109/MEMSYS.2019.8870899.

e). N. K. Khaira, T. Singh and R. R. Mansour, "Monolithically Integrated RF MEMS-Based Variable Attenuator for Millimeter-Wave Applications," in IEEE Transactions on Microwave Theory and Techniques, vol. 67, no. 8, pp. 3251-3259, Aug. 2019, doi: 10.1109/TMTT.2019.2925798.

f). O. D. Gurbuz and G. M. Rebeiz, "A 1.6–2.3-GHz RF MEMS Reconfigurable Quadrature Coupler and Its Application to a 360-deg Reflective-Type Phase Shifter," in IEEE Transactions on Microwave Theory and Techniques, vol. 63, no. 2, pp. 414-421, Feb. 2015, doi: 10.1109/TMTT.2014.2379258.

g). H. Yang, A. Yahiaoui, H. Zareie, P. Blondy and G. M. Rebeiz, "Symmetric and Compact Single-Pole Multiple-Throw (SP7T, SP11T) RF MEMS Switches," in Journal of Microelectromechanical Systems, vol. 24, no. 3, pp. 685-695, June 2015, doi: 10.1109/JMEMS.2014.2344694.

h). X. Li, V. P. J. Chung, M. G. Guney, T. Mukherjee, G. K. Fedder and J. Paramesh, "A High Dynamic Range CMOS-MEMS Accelerometer Array with Drift Compensation and Fine-Grain Offset Compensation," 2019 IEEE Custom Integrated Circuits Conference (CICC), Austin, TX, USA, 2019, pp. 1-4, doi: 10.1109/CICC.2019.8780142.

i). T. Singh, N. K. Khaira and R. R. Mansour, "Monolithically Integrated Reconfigurable RF MEMS Based Impedance Tuner on SOI Substrate," 2019 IEEE MTT-S International Microwave Symposium (IMS), Boston, MA, USA, 2019, pp. 790-792, doi: 10.1109/MWSYM.2019.8701106.

j). A. J. Alazemi and G. M. Rebeiz, "A low-loss 1.4–2.1 GHz compact tunable three-pole filter with improved stopband rejection using RF-MEMS capacitors," 2016 IEEE MTT-S International Microwave Symposium (IMS), San Francisco, CA, 2016, pp. 1-4, doi: 10.1109/MWSYM.2016.7540054.

k). N. Belkadi, K. Nadaud, C. Hallépée, D. Passerieux and P. Blondy, "Zero-Level Packaged 5W CW RF-MEMS Switched Capacitors," 2018 48th European Microwave Conference (EuMC), Madrid, 2018, pp. 559-562, doi: 10.23919/EuMC.2018.8541766.

l). T. Singh and R. R. Mansour, "Modeling of Frequency Shift in RF-MEMS Switches Under Residual Stress Gradient," 2018 18th International Symposium on Antenna Technology and Applied Electromagnetics (ANTEM), Waterloo, ON, 2018, pp. 1-2, doi: 10.1109/ANTEM.2018.8572954.

m). C. Ko, B. H. Ku, R. Gaddi and G. M. Rebeiz, "A 1–1.2 GHz RF MEMS VCO with accurate noise characterization," 2015 IEEE MTT-S International Microwave Symposium, Phoenix, AZ, 2015, pp. 1-3, doi: 10.1109/MWSYM.2015.7166993.

n). N. Belkadi, K. Nadaud, C. Hallepee, D. Passerieux and P. Blondy, "Zero-Level Packaged RF-MEMS Switched Capacitors on Glass Substrates," in Journal of Microelectromechanical Systems, vol. 29, no. 1, pp. 109-116, Feb. 2020, doi: 10.1109/JMEMS.2019.2949949.

o). G. M. Rebeiz, "RF MEMS: Theory, Design, and Technology,". John Wiley & Sons, 2004.

p). T. Singh, A. Elhady, H. Jia, A. Mojdeh, C. Kaplan, V. Sharma, M. Basha, E. Abdel-Rahman, "Modeling of low-damping laterally actuated electrostatic MEMS," Mechatronics, vol. 52, pp. 1-6, 2018, doi: 10.1016/j.mechatronics.2018.03.009.

q). A. H. Zahr et al., "RF-MEMS Switches for Millimeter-Wave Applications," 2019 European Microwave Conference in Central Europe (EuMCE), Prague, Czech Republic, 2019, pp. 336-338.

r). N. K. Khaira, T. Singh and R. R. Mansour, "RF MEMS Based 60 GHz Variable Attenuator," 2018 IEEE MTT-S International Microwave Workshop Series on Advanced Materials and Processes for RF and THz Applications (IMWS-AMP), Ann Arbor, MI, 2018, pp. 1-3, doi: 10.1109/IMWS-AMP.2018.8457154.

Round 2

Reviewer 2 Report

No comments